# Prostacyclin Regulation of Allergic Inflammation

**DOI:** 10.3390/biomedicines10112862

**Published:** 2022-11-09

**Authors:** Kunj Patel, R. Stokes Peebles

**Affiliations:** 1Division of Allergy, Pulmonary, and Critical Care Medicine, Vanderbilt University Medical Center, Nashville, TN 37232-2650, USA; 2Department of Pathology, Microbiology, and Immunology, Vanderbilt University School of Medicine, Nashville, TN 37232-2650, USA; 3United States Department of Veterans Affairs, Nashville, TN 37232-2650, USA; 4T-1218 MCN, Vanderbilt University Medical Center, 1161 21st Avenue South, Nashville, TN 37232-2650, USA

**Keywords:** prostacyclin, allergic inflammation, airway remodeling, proinflammatory, anti-inflammatory, cytokines

## Abstract

Prostacyclin is a metabolic product of the cyclooxygenase pathway that is constitutively expressed and can be induced during inflammatory conditions. While prostacyclin and its analogs have historically been considered effective vasodilators and used in treating pulmonary hypertension, prostacyclin has demonstrated potent anti-inflammatory effects in animal models of allergic airway inflammation. In vitro studies reveal that prostacyclin directly inhibits type 2 cytokine production from CD4+ Th2 cells and ILC2 and reduces the ability of dendritic cells to generate Th2 cytokine production from CD4+ T cells in an antigen-specific manner. Thus, there is strong evidence that prostacyclin may be an additional therapeutic target for treating allergic inflammation and asthma in human subjects.

## 1. Introduction

Allergic inflammation arises from both innate and adaptive immune responses that are characterized by an increase in the production of type 2 cytokines, such as interleukin (IL)-4, IL-5, IL-9, and IL-13 [1,2]. These cytokines are important in the pathogenesis of allergic diseases such as asthma, atopic dermatitis, allergic rhinitis, and food allergy. IL-4 can cause isotype switching of B cells to IgE production and drives naïve CD4+ T cells to a Th2 phenotype. IL-5 is the most important cytokine in eosinophil differentiation, migration, and survival, which is significant because eosinophils are critical in allergic inflammation pathogenesis. IL-9 has important effects on mast cell development, while IL-13 is a central mediator of airway responsiveness and mucus production, both cardinal features of asthma. An increased understanding of allergen-induced inflammation over the past decade reveals that activated CD4+ Th2 cells alone are not solely responsible for propagating allergic disease. In addition to CD4+ Th2 cells, group 2 innate lymphoid cells (ILC2) also produce IL-5, IL-9, and IL-13 and, in special circumstances, may also secrete IL-4 [3]. Importantly, ILC2 produces IL-5 at a significantly greater level than even CD4+ Th2 cells [4]. In patients with severe asthma, Th1, Th17, and CD8+ cytotoxic lymphocytes and neutrophils may even be observed [5]. 

In this review, we will focus on the role of prostaglandin (PG)I_2_ in regulating the allergic inflammatory pathway detailed in the preceding paragraph. As we will discuss, endogenous PGI_2_ restrains allergen-induced inflammatory responses, and both in vivo and in vitro studies in mice suggest that exogenous PGI_2_ may be a target for the treatment of allergic diseases such as asthma. PGI_2_ is one of the five primary prostaglandins. Prostaglandins are eicosanoids, active lipid compounds that are produced through arachidonic acid metabolism via the cyclooxygenase (COX) pathway. The sequential metabolism of arachidonic acid leading to prostaglandin production may be constitutive or induced in response to pathophysiological conditions, such as inflammation [6]. Arachidonic acid is released from the nuclear and cytoplasmic plasma membrane via the action of phospholipase A_2_ (PLA_2_). COX-1 and COX-2 convert arachidonic acid to an unstable intermediary, PGG_2_, which can then be converted to PGH_2_ by COX and peroxidase activity [7]. Tissue-specific enzymes and isomerases convert PGH_2_ into the five primary prostanoids, which include PGE_2_, PGD_2_, PGF_2α_, PGI_2_, and thromboxane A_2_. Whether a specific prostaglandin is produced or not is dependent upon the expression of the synthase for that prostaglandin in that tissue. 

PGI_2_ is most commonly referred to as prostacyclin, and we will use this terminology in this review. Prostacyclin was discovered in 1976 by Vane and Moncada [8]. Prostacyclin is generated by PGI synthase (PGIS) converting PGH_2_ into PGI_2_, and PGIS is primarily, but not exclusively, localized in endothelial cells, where it is most abundantly expressed (Figure 1) [7]. Traditionally, prostacyclin has been recognized as an effective vasodilator within both systemic and pulmonary circulations, as it elicits smooth muscle relaxation and has been observed to have anti-platelet aggregatory effects [9]. These characteristics are what led to the rise of prostacyclin analogs, such as iloprost, trepostinil, and epoprostenol, being used clinically in the treatment of pulmonary arterial hypertension [10,11,12]. For the FDA-approved prostacyclin analogs, iloprost can be administered through either the inhaled or intravenous routes, while epoprostenol is administered intravenously. Treprostinil can be given subcutaneously, intravenously, or orally [13,14]. While beraprost and cicaprost are also potent prostacyclin analogs, neither has been approved by the FDA [13].

## 2. Prostacyclin Signaling and In Vivo Models That Suggest an Inability to Signal through the IP Receptor Augments Allergic Inflammation

The biological effects of prostacyclin are exerted by signaling through its primary receptor, IP, which is also known as PTGIR in humans. IP is a seven transmembrane spanning G protein-coupled receptor (GPCR) that, when activated, results in increased intracellular cyclic AMP (cAMP) [7,10]. The increase in cAMP levels that result from IP signaling activates protein kinase A (PKA), leading to the phosphorylation of other proteins leading to downstream vasodilatory effects [10]. In addition to prostacyclin signaling through IP, a peroxisome proliferator-activated nuclear receptor (PPAR) may also be stimulated to act as a transcription factor [15]. PPARs exist in multiple isoforms, and while prostacyclin can signal through PPAR𝛿 in mice, PPARγ can be activated by prostacyclin analogs [15,16]. The prostacyclin-IP signaling pathway has been the focus of multiple studies using IP knock-out (KO) mice in order to gain a greater understanding of how prostacyclin regulates the allergic airway inflammatory response (Figure 2); however, as described above, IP is not the sole signaling pathway responsible for the effects of prostacyclin. 

An important tool in understanding how endogenous prostacyclin signaling regulates allergic airway inflammation was the creation of the IP deficient, or knock-out (KO), mouse. Early studies first detailed the effects of the IP receptor signaling in allergic airway inflammation by sensitizing and airway-challenging IP KO mice with ovalbumin to elicit a pulmonary allergen-driven inflammatory response. These studies reported a multi-fold increase in the number of leukocytes in bronchoalveolar lavage fluid (BALF) in IP KO mice relative to their wild-type (WT) counterparts [17,18]. In addition, IP KO mice had a significant increase in the eosinophilic airway and lung inflammation compared to similarly ovalbumin-sensitized and -challenged WT mice. In the ovalbumin model, IP KO mice had more than five times the amount of IL-4 and IL-5 produced in bronchoalveolar lavage fluid (BALF) relative to WT mice [17]. Furthermore, there was also a three-fold increase in the amount of interferon (IFN)-γ produced by splenic CD4+ T cells within the IP KO mice when compared to WT mice [17]. The restraining effect of IP signaling on the production of IFN-γ has since been corroborated by a more recent study that also noted an increase in the level of IFN-γ production from Natural Killer (NK) cells in IP KO mice challenged with house dust mice allergen compared to WT mice [19]. 

## 3. Dendritic Cells

Myeloid dendritic cells (mDC) are monocyte-derived antigen-presenting cells that are vital for initiating and regulating the adaptive immune response [20]. More specifically, mDCs accumulate in airways that are challenged by allergens and are involved in inducing and maintaining inflammatory responses by activating naïve T cells in the secondary lymphoid organs, implicating mDCs in potentially having a crucial role in allergic inflammation [20].

In a study that determined the effects of prostacyclin on dendritic cell function, ovalbumin-sensitized and challenged mice treated with inhaled iloprost in vivo had decreased expression of IL-4, IL-5, and IL-13 in restimulated lymph node cells compared to mice treated with the iloprost vehicle [11]. Iloprost treatment significantly decreased the number of eosinophils and lymphocytes in the BAL compartment and reduced peribronchial inflammation and goblet cell hyperplasia compared to vehicle treatment [11]. In vitro, activated iloprost-treated mouse dendritic cells that expressed CC chemokine receptor 7 (CCR7) had decreased chemotactic responsiveness to CC chemokine ligand 19 (CCL19) compared to vehicle-treatment, perhaps explaining the effect of iloprost treatment on the reduction of dendritic cell migration to the lymph nodes in vivo [11]. 

Another group reported that iloprost dose-dependently decreased TNF-α, IL-6, and IL-8 secretion by human mDCs compared to vehicle treatment [21]. A second group reported that, compared to vehicle treatment, iloprost decreased IL-6, IL-12, IL-23, and TNF-α, while it increased IL-10 in bone marrow dendritic cells (BMDCs) that were stimulated with ovalbumin and then intratracheally adoptively transferred into mice that were subsequently ovalbumin challenged through the airway [9]. Further, mouse CD4+ T cells that expressed a T cell receptor specific for an ovalbumin peptide had reduced production of IL-4, IL-5, and IL-13 when these cells were stimulated with DCs that had been cultured with ovalbumin BMDCs in the presence of iloprost compared to vehicle-treated BMDCs [11]. These results show promise for the use of prostacyclin analogs in attenuating adaptive allergic inflammatory responses.

The results of experiments detailing the effect of prostacyclin in downregulating proinflammatory cytokine production by mouse DCs are similar to those in which human DCs have been studied. Iloprost and treprostinil suppressed TNF-α expression by mDCs activated by the toll-like receptor (TLR) agonist, poly I:C. TNF-α is an important proinflammatory cytokine that recruits immune cells, regulates chemokine production, releases histamine, upregulates adhesion molecules, and is potentially involved in airway remodeling in dendritic cells [13,20]. Compared to vehicle treatment, iloprost-treated human mDCs also decreased the production of IFN-γ by CD4+ helper T cells, while iloprost enhanced the production of the immunosuppressive cytokine IL-10 [20]. These effects were seemingly modulated through the IP and PGE_2_ (EP) receptors but not PPARs [20]. Furthermore, the modulatory effects of treprostinil and iloprost in this study in which the cAMP pathway was activated were not completely IP-specific [20]. For instance, iloprost increased intracellular Ca2+ levels through EP1 receptor signaling and partly increased IL-10 levels while decreasing TNF-α via the EP1-Ca2+ pathway. In these studies, iloprost’s suppressive effects on TNF-α in human mDCs were a result of MAPK-p38-ATF2 pathway signaling [20]. Thus, some prostacyclin analogs, particularly at higher concentrations, may have IP-independent effects by activating EP receptors. Additionally, the study also examined the in vitro effects of prostacyclin analogs on epigenetic regulation. Epigenetic regulation, observed by the activity of histone acetyltransferase and deacetylase, in this instance, regulates inflammatory gene expression [20]. In patients with asthma, there is an overexpression in inflammatory genes due to a decrease in histone deacetylase activity and an increase in histone acetyltransferase activity, and these changes in acetylation through epigenetic regulation also regulate the proliferation and differentiation of T lymphocytes [20,22]. In mDCs that were activated by poly I:C, iloprost downregulated H3K4 trimethylation of the TNFA gene promoter region and inhibited poly I: C-induced translocation of methyltransferases [20].

### 3.1. Monocytes

In vitro studies also suggest that prostacyclin decreased cytokine secretion by human monocytes. Beraprost, iloprost, and treprostinil dose-dependently suppressed TNF-α expression, and iloprost was the most efficient of the three analogs based on the concentration needed to achieve a partial reduction in cytokine production [13]. These results provide a potential role for prostacyclin in helping to control the symptoms of asthma that may be due to increased TNF-α levels. The effects of prostacyclin in restraining cytokine production in monocytes may not be IP-dependent, as is the case in studies in which DCs were used. For example, in a study that analyzed the effects of prostacyclin analogs on both Th1 and Th2 cytokines in human monocytes, prostacyclin analogs suppressed Th1-related chemokine expression via PPAR-γ [23]. Additionally, the same study reported that iloprost and treprostinil suppressed IP-10, a Th1-related chemokine protein, via the IP-receptor-cAMP pathway [23]. Overall, these studies reflect the importance of better understanding the multiple pathways by which prostacyclin acts, as they can provide several avenues in which to study prostacyclin analog efficacy.

### 3.2. CD4+ Th2 Cells

Naïve CD4+ T cell differentiation into the Th2 subset is implicated as a key driver of allergen-induced inflammatory diseases such as asthma, and multiple studies reveal that prostacyclin inhibits Th2 inflammation [24]. For instance, IP KO mice sensitized and challenged with an extract of the ubiquitous aeroallergen *Alternaria* had significantly greater numbers of IL-5+ and IL-13+ CD4+ T cells compared to WT mice, further solidifying that endogenous IP signaling is a critical restraining influence on CD4+ Th2 cell differentiation [24]. While IL-4 is recognized as an important cytokine in differentiating CD4+ T cells down the Th2 pathway, IL-33 also has this effect. Cicaprost dose-dependently decreased IL-33-induced production of IL-4, IL-5, and IL-13 by CD4+ cells from WT mice but not in IP KO CD4+ T cells [24]. Cicaprost’s inability to suppress Th2 cytokine production in CD4+ Th2 cells from the IP KO mice confirmed the specificity of its effect on the prostacyclin-IP signaling pathway. Additionally, cicaprost decreased the IL-33-induced CD4+ Th2 cell production of IL-2 [24]. Prostacyclin reducing IL-2 could explain the decreased activation of the Th2 cells and their reduction in IL-5 and IL-13 production.

Signal Transducer and Activator of Transcription (STAT) 6 is a transcription factor that is activated by IL-4 and IL-13 and is critical for Th2 cell differentiation [25]. Indomethacin is a COX inhibitor that increases allergic proinflammatory cytokine responses in a STAT6-independent fashion [25]. Indomethacin administration likely resulted in increased allergic inflammation as a result of its decreasing prostacyclin production [25]. This was confirmed by a study using WT, STAT6 KO, IP KO, and IP-STAT6 double knock-out (DKO) mice. In this in vivo study in which ovalbumin was used to sensitize and challenge mice, IP KO mice had greater allergic lung inflammation compared to WT mice. STAT6 KO mice had undetectable levels of Th2 cytokines, while IP-STAT6 DKO mice also had significantly increased IL-5, IL-13, IL-1α, and IL-β protein expression compared to STAT6 KO mice [25]. This revealed that endogenous IP signaling inhibits a STAT6-independent pathway that can drive allergic inflammation [25]. 

### 3.3. Innate Allergic Inflammation and Group 2 Innate Lymphoid Cells (ILC2s)

ILC2s are tissue-resident cells that do not express T-cell, B-cell, or monocytic markers; however, they are able to produce Th2 cytokines, namely IL-5 and IL-13, at a markedly higher level than CD4+ Th2 cells [3]. ILC2s are instrumental in amplifying allergic inflammation in the lung [2]. 

In order to determine the effect of endogenous IP signaling on ILC2 function, mice were challenged with 4 consecutive days of *Alternaria* extract (Alt Ex) to elicit inflammation by innate immunity that precedes significant contribution from the adaptive immune response. In Alt Ex-challenged IP KO mice, there was an increase in lung IL-5 and IL-13 expression, as well as heightened airway eosinophilia and mucus compared to Alt Ex-challenged WT mice [2]. This revealed that endogenous IP signaling restrains ILC2-induced inflammation. In addition to these in vivo findings, in vitro data revealed that cicaprost inhibited IL-5 and IL-13 protein expression by IL-33-stimulated mouse ILC2s [2]. Likewise, in human ILC2s, cicaprost dose-dependently decreased IL-33-induced IL-5 and IL-13 production [2]. Others have reported similar findings. In a study examining mice stimulated with intranasal IL-33, the iloprost-treated group had lower mRNA expression levels of IL-5 and IL-13 alongside a reduction in the proliferation of ILC2s relative to mice treated with the vehicle for iloprost, demonstrating the attenuating effects of prostacyclin analogs on ILC2 type 2 cytokine production [26]. These findings were supported by another study that reported that iloprost directly suppressed ILC2 cytokine production following IL-33 activation [27].

### 3.4. Th17 Cells

Th17 cells are a subset of CD4+ helper T cells that produce the cytokines IL-17A, IL-17F, IL-21, and IL-22. The differentiation of naïve CD4+ T cells into Th17 cells requires the expression of TGF-β and IL-6 in mice, or TGF-β, IL-1 β, and IL-21 in humans [28]. When IL-17A and IL-17F are produced from Th17 cells, they act in a proinflammatory role and can induce other proinflammatory cytokines, such as IL-8, from the airway epithelium, resulting in neutrophil recruitment. IL-17A and IL-17F also can elicit smooth muscle contraction and airway mucus metaplasia. Both cytokines are increased in the airways of patients with severe asthma [28].

Eicosanoids clearly regulate the production of Th17 cytokines. COX-2 KO mice had significantly reduced numbers of Th17 cells in the lungs, BALF, and spleen following ovalbumin-sensitization and challenge compared to WT mice [28]. Further, IL-17A and IL-6 levels in the blood and BALF were markedly reduced in COX-2 KO mice relative to WT mice following the allergen challenge [28]. These investigators also examined the roles of several different prostaglandins in Th17 differentiation and demonstrated that prostacyclin increased Th17 differentiation in cells that express COX-2 and partially restored Th17 differentiation in mice that were deficient for COX-2 [28]. This finding was corroborated by another group that reported that when iloprost and cicaprost administration in the presence of IL-23 promoted Th17 differentiation and survival in vitro [29]. When prostacyclin analogs were administered in vitro to naïve CD4+ T cells from WT mice, they stimulated greater production of both IL-17A and IL-22 compared to vehicle treatment, indicating that the analogs supported Th17 differentiation and revealing another facet in prostacyclin’s role in allergic inflammation [29].

γδ T cells have a first-line immunoprotective role against pathogens and environmental irritants in mucosal tissues [30]. A subset of γδ T cells, γδ-17 cells, produce IL-17. IP KO mice challenged by ovalbumin had a significant reduction in the amount of γδ-17 cells compared to WT mice, and iloprost augmented IL-17 production by γδ T cells, reflecting that both prostacyclin analogs and prostacyclin signaling play a key role in γδ T cell development and effector function [30].

### 3.5. Treg Cells

Regulatory T cells (Tregs) are critical for immune tolerance in that they suppress inflammation, promote tolerance, and inhibit the development of autoimmune disorders [9,31]. Development of Tregs is promoted by the transcription factor Forkhead box p3 (Foxp3) [31]. Regulatory T cells are crucial for suppressing type 2 inflammation and do so through expressing GATA-binding protein 3 (GATA3) and interferon regulatory factor 4 (IRF4) in order to inhibit Th2 cells [31]. Tregs that express the inhibitory receptor immunoglobulin-like transcript 3 (ILT3) are unable to suppress the Th2 response [31].

In vivo studies reveal that prostacyclin signaling promotes immune tolerance and Treg stability. In the ovalbumin sensitization and challenge model, immune tolerance can be induced by exposing WT mice to aerosolized ovalbumin prior to sensitization. IP KO mice did not develop tolerance in this protocol, suggesting that IP signaling was critical for tolerance induction [32]. Interestingly, in this study, a surprising finding was that there were more, not fewer, Treg in the lungs of IP KO mice. This result suggested that Treg in IP KO mice may be less functional than those in WT mice [32]. A follow-up study confirmed that IP signaling was critical for Treg effector function. Treg from IP KO mice had decreased Foxp3 expression compared to WT Treg, and Foxp3 expression is correlated with Treg suppressive function [31]. In an in vivo adoptive transfer model, Treg from IP KO mice was significantly less able to suppress allergic inflammation compared to WT Treg [31]. Lung IP KO Tregs isolated from mice had a significantly greater ILT3, revealing that prostacyclin signaling attenuated ILT3 expression on Tregs. Further, in vitro studies showed that cicaprost promoted Foxp3 expression in both mouse and human T cells polarized down the Treg pathway. Another group reported that in the ovalbumin-induced asthma model, iloprost promoted regulatory T cell differentiation from naïve T cells [9]. Thus, prostacyclin signaling is critical for optimal Treg function.

### 3.6. NK Cells

Natural Killer (NK) cells are cells of the innate immune system that serve to target and eliminate tumors and cells infected by viruses [19]. There have been conflicting reports on how exactly NK cells modulate allergic lung inflammation, where some have reported that NK cells play an inhibitory role, and others report that NK cells promote allergen-induced inflammation [19]. One study reported the role of prostacyclin signaling in NK cell function in allergic airway inflammation and found that IP KO mice had reduced allergic lung inflammation induced by house dust mites and decreased Th2 cytokine production [19]. While this finding is in contrast to data previously mentioned in this review, in this model, IP KO mice had a greater number of pulmonary NK cells than WT mice. Depleting NK cells restored allergic inflammation in the IP KO mice to levels seen in WT mice, and transferring NK cells into airways suppressed allergic inflammation [19]. Thus, this leads to the speculation that IP signaling promotes NK cell function to regulate allergies.

## 4. In Vivo Models that Support the Use of Prostacyclin in Reducing Airway Remodeling and Asthmatic Symptoms

The synthetic prostacyclin analog ONO-1301 reduced allergic inflammation, airway hyperresponsiveness, and remodeling in mice in the ovalbumin model [33]. Mice administered ONO-1301 had decreased goblet-cell metaplasia, reduced airway smooth muscle hypertrophy, and inhibited submucosal collagen deposition [33]. These results support the possibility that prostacyclin may be a potential therapeutic approach to reduce airway remodeling [18,34].

The relationship between cough reflex sensitivity and airway inflammation was investigated by observing the effect of the prostacyclin analog beraprost on asthmatic patients. Unfortunately, beraprost decreased the cough threshold and thus enhanced cough reflex sensitivity in the subjects with asthma [35]. Contrary to these findings, a more recent study examined the role of prostacyclin in the cough response by triggering bronchoconstriction via methacholine chloride (MCh) inhalation in guinea pigs [36]. In animals that were administered a high dose of prostacyclin, the number of coughs induced by bronchoconstriction was significantly decreased, and when an IP antagonist was incorporated, the number of coughs increased [36]. Thus, while there are conflicting reports on the exact effects of prostacyclin on the cough response, there is potential for prostacyclin analogs to reduce a nearly ubiquitous symptom of asthma and allergic inflammation.

## 5. Concluding Remarks

The landscape of asthma treatment has changed drastically over the last decade with the FDA approval of biologics that target specific molecules that contribute to allergic inflammation. Antibodies against IgE, IL-5, the IL-4 receptor α, and TSLP are all clinically efficacious. However, there remain patients who have uncontrolled asthma despite adequate trials of biological therapies, and this may be where prostacyclin may be useful. Future steps towards potentially utilizing prostacyclin analogs for allergic inflammation, particularly asthma, would require clinical trials. A study that examined the feasibility of administering inhaled iloprost in human patients with mild asthma found that this prostacyclin analog was safe [37]. While there are still aspects to learn about the mechanism for how prostacyclin signaling reduces allergic inflammatory responses, the published data in this review support further research into repurposing prostacyclin as a possible treatment for allergic inflammation and asthma.

## Figures and Tables

**Figure 1 biomedicines-10-02862-f001:**
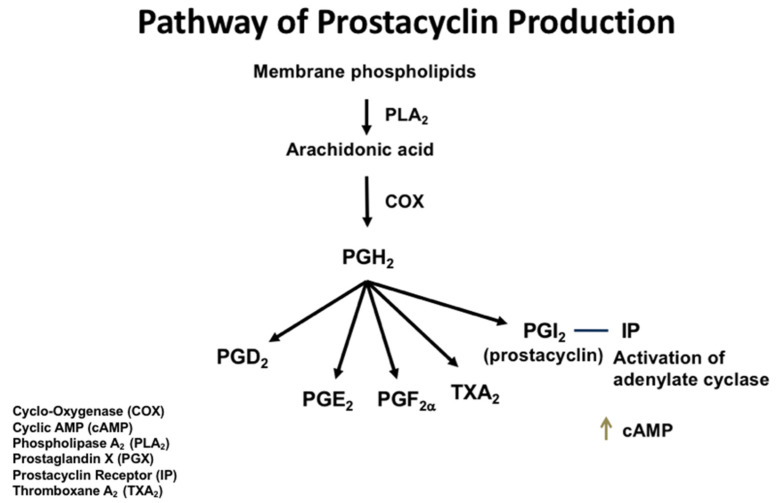
The pathway of prostacyclin production. The arrows show the downstream enzymatic pathways.

**Figure 2 biomedicines-10-02862-f002:**
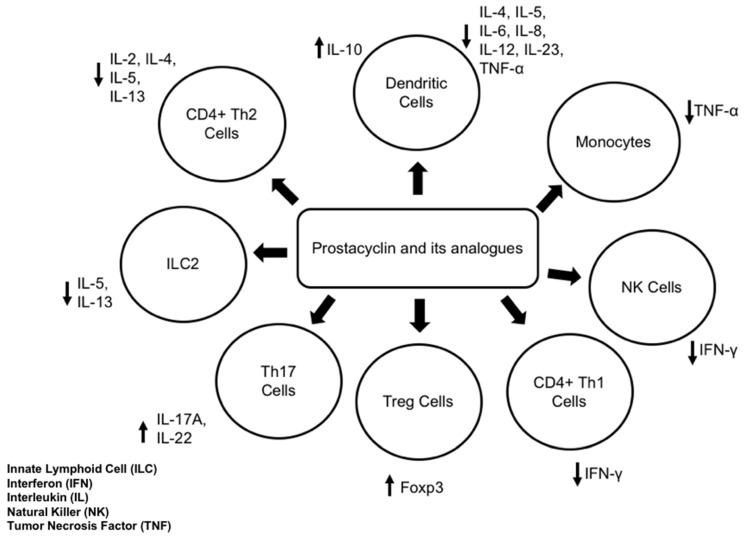
The effects of prostacyclin and its analogs on functions of leukocytes that are important in allergic inflammatory responses. The arrows show how prostacyclin and its analogues regulate specific cell types.

## Data Availability

Not applicable.

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
