# Peer review of "Prostacyclin Regulation of Allergic Inflammation"

_biomedicines, 2022, doi:10.3390/biomedicines10112862_

Round 1

Reviewer 1 Report

A well-written paper reviewing the current knowledge on the role of prostacyclin in Allergic Inflammation. Although the authors investigated the close relationship between prostacyclin and mediators of the immune system, they did not add any information on the cross-talk between prostacyclin and neurotrophins in affecting the pathogenesis and the development of allergic inflammation. Recently, this field has been highly investigated and is worthy of attention. Please, go in deep on this issue, also referring to the following papers:

1. The Role of Neurotrophins in Inflammation and Allergy. Vitam Horm. 2017;104:313-341. doi: 10.1016/bs.vh.2016.10.010

2. International Union of Basic and Clinical Pharmacology. CIX. Differences and Similarities between Human and Rodent Prostaglandin E2 Receptors (EP1–4) and Prostacyclin Receptor (IP): Specific Roles in Pathophysiologic Conditions

Reviewer 2 Report

With interest, I read the manuscript biomedicines-1972289. Overall, it is a nice, compact mini-review addressing an important yet not sufficiently attended topic. Very nice manuscript.

Normally, I would consider this draft to short but in this very special case I do not. It is because it is compact but very reach in information. Besides, the topic is innovative and not much is known yet. Third, it is up to the Editors to decide on things like that.

I have some minor comments only:

1.       The legends to the figures should be elaborated, e.g. all abbreviations should be explained.

2.       The role of epigenetic regulation in regulating maturation and differentiation of white blood cells, especially T lymphocytes could be mentioned in 1-2 sentences (facultative).
